# Dietary Behaviors and Incident COVID-19 in the UK Biobank

**DOI:** 10.3390/nu13062114

**Published:** 2021-06-20

**Authors:** Thanh-Huyen T. Vu, Kelsey J. Rydland, Chad J. Achenbach, Linda Van Horn, Marilyn C. Cornelis

**Affiliations:** 1Department of Preventive Medicine, Feinberg School of Medicine, Northwestern University, Chicago, IL 60611, USA; huyenvu@northwestern.edu (T.-H.T.V.); c-achenbach@northwestern.edu (C.J.A.); lvanhorn@northwestern.edu (L.V.H.); 2Research and Information Services, Feinberg School of Medicine, Northwestern University, Chicago, IL 60611, USA; kelsey.rydland@northwestern.edu; 3Department of Medicine, Feinberg School of Medicine, Northwestern University, Chicago, IL 60611, USA

**Keywords:** nutrition, COVID-19, immunity, coffee, breastfeeding, epidemiology

## Abstract

Background: Nutritional status influences immunity but its specific association with susceptibility to COVID-19 remains unclear. We examined the association of specific dietary data and incident COVID-19 in the UK Biobank (UKB). Methods: We considered UKB participants in England with self-reported baseline (2006–2010) data and linked them to Public Health England COVID-19 test results—performed on samples from combined nose/throat swabs, using real time polymerase chain reaction (RT-PCR)—between March and November 2020. Baseline diet factors included breastfed as baby and specific consumption of coffee, tea, oily fish, processed meat, red meat, fruit, and vegetables. Individual COVID-19 exposure was estimated using the UK’s average monthly positive case rate per specific geo-populations. Logistic regression estimated the odds of COVID-19 positivity by diet status adjusting for baseline socio-demographic factors, medical history, and other lifestyle factors. Another model was further adjusted for COVID-19 exposure. Results: Eligible UKB participants (*n* = 37,988) were 40 to 70 years of age at baseline; 17% tested positive for COVID-19 by SAR-CoV-2 PCR. After multivariable adjustment, the odds (95% CI) of COVID-19 positivity was 0.90 (0.83, 0.96) when consuming 2–3 cups of coffee/day (vs. <1 cup/day), 0.88 (0.80, 0.98) when consuming vegetables in the third quartile of servings/day (vs. lowest quartile), 1.14 (1.01, 1.29) when consuming fourth quartile servings of processed meats (vs. lowest quartile), and 0.91 (0.85, 0.98) when having been breastfed (vs. not breastfed). Associations were attenuated when further adjusted for COVID-19 exposure, but patterns of associations remained. Conclusions: In the UK Biobank, consumption of coffee, vegetables, and being breastfed as a baby were favorably associated with incident COVID-19; intake of processed meat was adversely associated. Although these findings warrant independent confirmation, adherence to certain dietary behaviors may be an additional tool to existing COVID-19 protection guidelines to limit the spread of this virus.

## 1. Introduction

Coronavirus (COVID-19) is an infectious disease caused by SARS-CoV-2 that predominantly attacks the respiratory system and has spread rapidly in the last year [1]. This unprecedented global public health crisis necessitates the expansion of our understanding of COVID-19 for purposes of prevention and mitigation of its severity and mortality risk in infected individuals [2].

Cross-sectional analyses identify patient characteristics and clinical data subsequent to a positive test, thereby informing the progression and severity of COVID-19. However, identifying factors underlying potential susceptibility a priori to the virus offers public health benefits. The immune system plays a key role in an individual’s susceptibility and response to infectious diseases, including COVID-19 [3,4]. A major modifiable factor affecting immune function is dietary behavior that influences nutritional status [5,6,7]. Ecological studies of COVID-19 report favorable correlations with specific vegetables and dietary patterns, such as the Mediterranean diet [8,9,10]. Some dietary supplements were found to have an association with SARS-CoV-2 infection [11]. To our knowledge, no population data have been examined regarding the role of specific dietary intakes in prevention of COVID-19.

Using data from the UK Biobank (UKB), we examined the associations between dietary behaviors measured in 2006–2010 and incident COVID-19 infection in 2020. We additionally linked UKB geo-data to UK COVID-19 surveillance data to account for COVID-19 exposure in our analyses, which to our knowledge has never been done in COVID-19 studies of the UKB.

## 2. Materials and Methods

### 2.1. UK Biobank

The UKB is an international health resource of over 500,000 participants aged 37–73 years at 22 centers across England, Wales, and Scotland. Details of the study methods have been described previously [12]. Briefly, participants underwent physical measurements, assessments about health and risk factors (including lifestyle and dietary behaviors), and blood sampling at baseline (2006–2010) and agreed to follow-up on their health status. Details of data collected are provided on the Showcase tab of the UKB website [13]. UKB ethical approval was from the National Research Ethics Service Committee North West–Haydock (approval letter dated 17 June 2011, Ref 11/NW/0382), and all study procedures were performed in accordance with the World Medical Association Declaration of Helsinki ethical principles for medical research. The current analysis was approved under the UKB application #21394 (PI, M.C.C).

### 2.2. COVID-19 Diagnosis

COVID-19 test results from Public Health England have been dynamically linked to the UKB since 16 March 2020 [14]. The regularly updated COVID-19 data table provided to UKB researchers includes participant ID, record date, test location (mouth, nose, throat, trachea, etc.), testing laboratory (71 labs listed), and test results (negative or positive). The vast majority of samples tested are from combined nose/throat swabs that are transported in a medium suitable for viruses and subject to polymerase chain reaction testing.

### 2.3. COVID-19 Exposure

UKB provides grid co-ordinate data at the 1 km resolution as a measure of residential location [15]. The data are provided to researchers in the projection of the Ordnance Survey National Grid (OSGB1936) geographic reference system; measures refer to easting and northing with a reference point near the Isles of Sicily. These data were used in conjunction with country-wide surveillance data to identify UKB participants exposed to COVID-19 [16]. The UKB geo-data and UK COVID-19 surveillance data was imported and projected in ArcGIS (Environmental Systems Research Institute, Redlands, CA, USA) for visual inspection. To make the UKB geo-data (North/East) compatible with the latitude/longitude surveillance data, we converted the former to the latter using standard equations already built into ArcGIS [17]. Hotspot analysis was performed on the UK surveillance data to assign values (“Gi*” statistic) back to each UKB participant representing a COVID-19-exposure risk factor or score. We calculated monthly positive case rates per specific geo-populations, and the average of these monthly rates was used in the analysis.

### 2.4. Baseline Dietary Data

The UKB touchscreen at baseline included a semi-quantitative food frequency questionnaire (FFQ) that assessed self-reported usual intake of 17 foods and beverages [18]. Participants were asked to report the number of pieces/tablespoons/cups of each item they consumed or to choose one of several pre-specified frequency categories. Among these choices, baseline servings of vegetables (cooked, raw), fruit (fresh, dried), oily fish, processed meat, red meat (beef, lamb/mutton, or pork), tea, and coffee have specifically been identified as contributing nutritional factors implicated in immunity [19,20,21,22,23,24] and were thus targeted for consideration in this study. The details of the questions and possible answers for these diet factors are provided in Appendix A. Based on the participants’ responses, we computed portion sizes as cups or servings per day. Participants were also asked if they were breastfed as a baby and responded yes, no, or don’t know.

### 2.5. Other Covariates

Age, sex, race, education, employment status, type of accommodation lived in and number of co-habitants, smoking behaviors, and current health status were self-reported at baseline using the touchscreen. Townsend Deprivation Index, a measure of SES, was derived using census data and postal codes of participants, where higher scores represent higher deprivation. Body mass index (BMI) was calculated using height and weight measured at the assessment center. Physical activity was assessed using questions adapted from the validated short International Physical Activity Questionnaire [25]. Participants were asked how many minutes they engaged in each of the activities on a typical day. Duration of moderate and vigorous activities (minutes/day) was used in this analysis. History of diabetes, history of any heart diseases, and hypercholesterolemia or hypertensive medication use were obtained via self-reporting (and verified during a verbal interview) and hospital databases.

Because individuals with lung diseases may have an increased risk of COVID-19 infection [26], we further classified participants by their pre-existing compromised pulmonary function (yes/no respiratory diseases excluding common colds); data extracted from health records during follow-up. We also used health records to calculate incident diabetes and heart disease.

### 2.6. Analysis Sample

Because no COVID-19 test data were available for UKB assessment centers in Scotland and Wales, only participants in England were included. For this analysis, we included participants with test results between 16 March and 30 November 2020, which was before vaccines were rolled out in the UK (8 December 2020) [27]. There were 74,878 tests performed on 42,699 UKB participants. We excluded participants who had missing baseline data on nutritional factors and covariates (*n* = 4711). The final study sample included 37,988 UKB participants (Appendix A).

### 2.7. Statistical Analysis

All analyses were performed using SAS (SAS Institute, Inc., Cary, NC, USA). Mapping was performed using ArcGIS. F-tests or χ^2^ tests were used to compare differences in baseline characteristics across participants’ age and race/ethnicity.

Our main outcome of interest was whether a person had any confirmed COVID-19 infection (defined as having any positive test result). Our exposures of interest included consumption of coffee, tea, oily fish, processed meat, red meat, fruit, and vegetables; and were analyzed in quartiles of servings/d [each modeled: lowest quartile/Q1(referent), Q2, Q3, and Q4] as well as tea and coffee [each modeled: none or <1 (referent), 1, 2–3, and ≥4 cups/d], and breastfed as a baby [yes (referent), no, and don’t know]. Because the cut-points to define Q2 and Q3 for oily fish were similar, Q3 and Q4 were combined.

We examined the association between each dietary behavior and incident COVID-19 using logistic regression, first adjusting for age, sex, and race (White/Asian/Black/Mixed-Others) (Crude model), then further adjusting for Townsend-deprivation index (quartiles), education (6 qualification classes), employment status (employed/retired/other), type of living accommodation (house/apartment/other), number of co-habitants (1,2,3 or ≥4), income (4 levels), physical activity (quartiles of moderate or vigorous activities, min/day), smoking (never/past/current), BMI (<25, 25–<30, and ≥30 kg/m^2^), self-rated health (excellent/good/fair/poor), hypercholesterolemia medication use (yes/no), hypertensive medication use (yes/no), diabetes (yes/no), and any heart diseases (yes/no) (Model 1). Model 2 was similar to Model 1 but with diet behaviors assessed mutually (i.e., including all 7 other dietary factors). Model 3 included variables from Model 2 and further adjusted for COVID-19 exposure score. Finally, Model 4 further adjusted for pre-existing compromised pulmonary function (yes/no). Statistical significance was defined as *p* < 0.05. No adjustments were made for multiple testing as all tests were a priori.

In sensitivity analyses, we excluded individuals self-reporting diabetes, heart diseases, or using cholesterol-lowering or antihypertension medication because they may recently have made dietary changes consequential to the disease. We also adjusted for incident diabetes and heart disease during follow-up. Moreover, as shown in Appendix A, there were two waves of new cases between March and November 2020 (i.e., from March–August 2020, or Wave 1, and September–November 2020, or Wave 2) in the UK, with greater numbers in Wave 2 than Wave 1. The differences may reflect a change in testing-rate, a true change in risk of exposure to the virus or some other factor. Regardless, analyses were also conducted for Wave 1 and Wave 2 separately.

We screened for effect modification (interaction) by sex (male or female), age (<55 or ≥55 years of age) and race (White, Asian, Black, or mixed/others) by including the cross-product term of each nutritional factor (e.g., coffee consumption, 4 categories) and the interacting variable in multivariable regression models. Statistically significant interactions were defined as *p* < 0.002; accounting for 24 tests (8 nutrition factors × 3 effect modifiers) performed.

## 3. Results

### 3.1. Participant Characteristics

Of 37,988 eligible participants who were tested for COVID-19 from March to November 2020, 6482 (17%) tested positive. In general, participants in the analysis sample had baseline demographic and diet characteristics similar to those of the full UKB cohort (*N* = 502,633, Table 1). However, those in the analysis sample tended to report poorer health and more comorbidities than those in the full UKB cohort.

Characteristics of the analysis sample, stratified by baseline age and race, are presented in Appendix A. Briefly, compared to the older age group, the younger age group tended to be female, employed, and better educated, with better income and rated better health; they also tended to consume less tea, fruit, vegetables, fish, and red meat, with fewer having been breastfed. Compared to non-white participants, white participants tended to consume more coffee, tea, processed meat, less fruit or vegetables, with fewer having been breastfed. Among non-whites, Black people tended to live in higher deprivation areas, be employed, have higher BMI, and consume more red meat, while Asians tended to report poorer health and consume more fruit and vegetables. The Asian group also had a higher incidence of COVID-19 positivity than other racial groups.

The exposure risk for the UKB sample in the context of the nation overall is illustrated in Figure 1 and Appendix A, in which geo-positive tests are shown by Waves for UKB and UK data separately. Areas with a high number of positive tests in UKB were generally similar to those in the UK overall.

### 3.2. Nutritional Factors and COVID-19 Positivity

With adjustment for age, race, and sex (Crude model), consumption of coffee, moderate tea, oily fish, and vegetables; and being breastfed as a baby were significantly associated with lower odds of COVID-19 positivity, while consuming processed meat was associated with higher odds of COVID-19 positivity (Appendix A). After further adjusting for other socio-demographics, medical and lifestyle factors, associations were attenuated; consumption of tea and fish oil were no longer significantly associated with COVID-19 infection (Model 1–Table 2). The odds (95% CI) of COVID-19 positivity were 0.90 (0.83, 0.98), 0.90 (0.83, 0.96), and 0.92 (0.84, 1.00) when consuming 1 cup, 2–3 cups, and 4+ cups of coffee/day (vs. <1 cup/day), respectively (Model 2). The odds (95%CI) of COVID-19 infection was 0.88 (0.80, 0.98) for individuals in the 3rd quartile of vegetable intake (vs. lowest quartile), 1.14 (1.01, 1.29) for individuals in the 4th quartile of processed meat intake (vs lowest quartile), and 0.91 (0.85, 0.98) for those who had been breastfed as a baby (vs. not breastfed). We observed similar associations between these dietary factors assessed individually (Model 1) or mutually (Model 2) with COVID-19 infection, suggesting independent associations. Associations were attenuated when further adjusted for COVID-19 exposure, but patterns of associations remained (Model 3). Further adjusting for pre-existing compromised pulmonary function, or incident diabetes and heart disease during follow-up did not alter results (Model 4, data not shown).

Patterns of associations were similar when excluding individuals reporting diabetes, CVD, or the use of cholesterol-lowering or antihypertension medication from the sample (data not shown), or when analyzing data for Wave 1 and Wave 2 separately (Appendix A). Associations between nutrition factors and COVID-19 positivity were not significantly modified by age, sex, and race (*p* > 0.002 for interactions, Appendix A).

## 4. Discussion

In this study, consuming more coffee, vegetables, and being breast fed as well as consuming less processed meat intake were independently associated with lower odds of COVID-19 positivity. These associations were attenuated when accounting for the UK’s COVID-19 case rate (i.e., exposure).

Much research has focused on characterizing individuals who test positive for COVID-19 and infection progression and outcomes. Individuals with suppressed immune systems, such as the elderly and those with existing comorbidities including cardiovascular diseases, hypertension, diabetes, and obesity, are more likely to suffer with progression toward severe outcomes of COVID-19 [3,4,28,29]. Less attention has focused on modifiable risk factors preceding COVID-19 infection. While nutrition may theoretically impact COVID-19 susceptibility [3,6,30,31,32], few investigations have specifically tested the hypothesis *a priori*. Low vitamin D status is associated with COVID-19 infection, severity, and mortality [33,34]. Del Ser et al. [35] recently explored a variety of risk factors for the incidence, severity, and mortality of COVID-19 in a Spanish cohort of 913 volunteers aged 75–90 years. Sixty-two cases reported symptoms compatible with COVID-19; 6 of them died. High alcohol and lower coffee and tea consumption were associated with disease severity; other dietary behaviors were not considered.

In the UKB, habitual consumption of 1 or more cups of coffee per day was associated with about a 10% decrease in risk of COVID-19 compared to less than 1 cup/day. Coffee is not only a key source of caffeine, but contributes dozens of other constituents; including many implicated in immunity [21]. Among many populations, coffee is the major contributor to total polyphenol intake, phenolic acids in particular [36,37]. Coffee, caffeine, and polyphenols have antioxidant and anti-inflammatory properties [38,39,40,41,42,43,44]. Coffee consumption favorably correlates with inflammatory biomarkers such as CRP, interleukin-6 (IL-6), and tumor necrosis factor α (TNF-α) [38,45,46,47,48,49,50,51], which are also associated with COVID-19 severity and mortality [52,53]. Coffee consumption has also been associated with lower risk of pneumonia in elderly [54]. Taken together, an immunoprotective effect of coffee against COVID-19 is plausible and merits further investigation.

Fruits and vegetables are rich dietary sources of vitamins, folate, fiber, and several phytochemicals such as carotenoids and flavonoids [55,56]. These substances have anti-inflammatory, antibacterial, and antiviral properties and are thus immune-protective [19,57,58]. In the current study, consumption of at least 0.67 servings/d of vegetables (cooked or raw, excluding potatoes) was associated with a lower risk of COVID-19 infection. Recent ecological studies of COVID-19 report that countries with high consumption of foods with potent antioxidant or anti angiotensin-converting enzyme (ACE) activity such as raw or fermented cabbage have a lower COVID-19 death rate compared to other countries [59,60]. Studies of vegetables and fermented-foods with COVID-19 mortality in Europe reported that each g/day increase in the average national consumption of head cabbage, cucumber or fermented vegetables decreased the mortality risk for COVID-19 by 11–35%; consumption of broccoli surprisingly increased COVID-19 mortality [9,10]. COVID-19 mortality was not our outcome of interest, and while we did not analyze vegetables individually, the UK’s national consumption of broccoli is above Europe’s average and thus our findings conflict with the previous inverse relationship between broccoli consumption and COVID-19 mortality [9,10]. Fruit (fresh and dried) consumption was not associated with COVID-19 risk in the UKB. While fruits and vegetables share several health benefits, the specific bioactive compounds in fruits and in vegetables can vary [61]. Fruits are also relatively higher in sugar (fructose) while vegetables contain more starch. Follow-up studies to our current findings might therefore focus on vegetable constituents distinct from those of fruit.

Processed meat consumption of as little as 0.43 servings/d was associated with a higher risk of COVID-19 in UKB. However red meat consumption presented no risk, suggesting meat *per se* does not underlie the association we observed with processed meats. Processed meat refers to any meat that has been transformed through salting, curing, fermenting, smoking, or other process to enhance flavor or improve preservation [62]. In the UK, sausages, bacon, and ham are the major contributors to processed meat intake [63] and these often contain salt enriched with nitrates/nitrites. Preservatives and other additives are increasingly being used yet difficult to measure in observational studies [64]. These include milk-, soy- and wheat-based ingredients, spices, ascorbic acid, phosphates, antioxidants, monosodium glutamate, food colorings, blood plasma, gelatin, and transglutaminase [62]. Total and saturated fat concentrations are also relatively higher in processed meats [65]. Processed meats are also characteristic of a western-style diet, which may adversely impact immunity [23] and thus other dietary behaviors that correlate with processed meat intake may underlie the association we observed with COVID-19 susceptibly.

Consumption of human milk has also emerged as an early-life factor impacting immunity in both infancy and adulthood, with links to lower rates of allergy, influenza, asthma, and other respiratory infections [66,67]. Breastfeeding has also been linked to epigenetic changes of toll-like-receptors, which play significant roles in innate immunity [68]. We found a long-term favorable association between being breastfed as a baby and COVID-19 infection in UKB and thus contribute to the growing evidence in support of nutrition early in life for optimal immunity for life.

The strengths of this study include access to the largest cohort established years before the COVID-19 pandemic, with detailed health, lifestyle, and nutrition data and ongoing follow-up. Although much COVID-19 research [69] has incorporated this database, we are the first to leverage independent COVID-19 surveillance database of the UK to estimate participant COVID-19 exposure [70]. COVID-19 data from UKB are consistent with public government data, which adds confidence to our approach and results.

The study has limitations. First, only a portion of overall UKB participants were tested for COVID-19 (about 10%) during the study timeframe, and these participants were slightly older, less educated, and less employed, while reporting poorer health than the original UKB cohort. Those factors were associated with higher odds of COVID-19 infection in our analysis sample. We note, however, that the full UKB is not representative of the sampling population, with evidence of a ‘healthy volunteer’ selection bias [71]. However, participants in these two samples had many baseline characteristics in common, including dietary behaviors. Any effects of selection bias would likely lead to an underestimation of the true associations between nutrition and COVID-19. Second, diet assessment tools are generally prone to measurement error: limitations inherent to closed-ended options, inaccurate portion size estimates, and social desirability bias are among the sources contributing to error [12,72]. Nevertheless, we were uniquely positioned to pursue this investigation with novel aims and analyses; and although the effect sizes may be imprecise, establishing a relationship between diet and COVID-19 is informative in itself and can be inferred from the direction and statistical significance of the association [12,72]. Finally, the current study is observational and cannot infer causality. While we accounted for a comprehensive set of potential confounders, we cannot discount the possibility of residual confounding; whether it be positive or negative. Moreover, we had no concurrent pandemic data on other established risk factors for COVID-19 infection, such as participant social distancing behavior, work environment, and face mask-use; some of these factors may correlate with diet behaviors.

## 5. Conclusions

Our results support the hypothesis that nutritional factors may influence distinct aspects of the immune system, hence susceptibility to COVID-19. Encouraging adherence to certain nutritional behaviors (e.g., increasing vegetable intake and reducing processed meat intake) may be an additional tool to existing COVID-19 protection guidelines to limit the spread of this virus. Nevertheless, our findings warrant confirmation in other populations.

## Figures and Tables

**Figure 1 nutrients-13-02114-f001:**
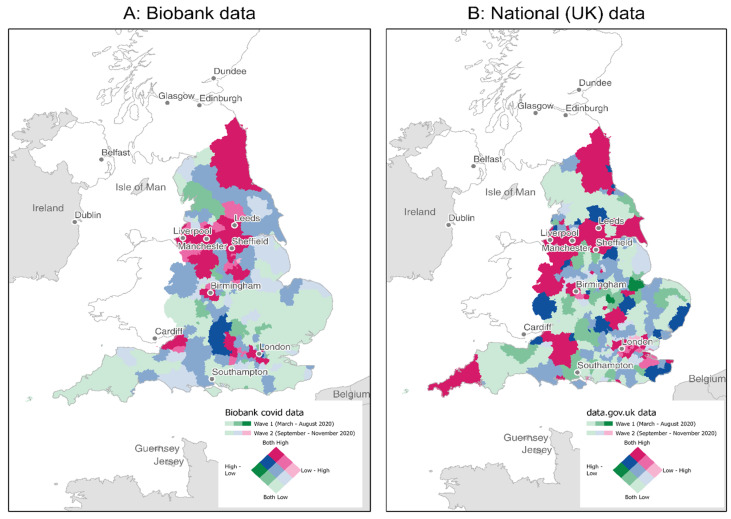
Total COVID-19 Positive Tests in Wave 1 compared to Wave 2 by UKB (**A**) and National (UK) (**B**) Data The UKB geo-data and National (UK) COVID-19 surveillance data was imported, projected, and converted to be compatible with each other in ArcGIS for visual inspection (see COVID-19 exposure in Materials and Methods section for more details). Wave 1: Total positive tests for March-August 2020; Wave 2: Total positive tests for September–November 2020; Source: UKB and https://coronavirus.data.gov.uk/ accessed on 4 April 2021.

**Table 1 nutrients-13-02114-t001:** Baseline characteristics of the analysis sample and the full UK Biobank cohort.

Baseline Characteristics *	Analysis Sample	Full UKB Cohort
Number of Persons	37,988	502,633
Age, yr, mean(sd)	57.36 (8.23)	56.53 (8.10)
Female	20,026 (52.72)	273,453 (54.41)
Townsend-deprivation index, mean(sd)	−1.21 (3.12)	−1.29 (3.10)
White/British	35,793 (94.22)	472,801 (94.06)
Household Income, £ < 18,000	8264 (21.75)	97,220 (19.34)
College or university degree	11,000 (28.96)	161,198 (32.07)
Currently employed	20,354 (53.58)	287,210 (57.14)
Homeowner	34,295 (90.28)	447,795 (89.36)
Number of co-habitants ≥ 4	6821 (17.96)	93,987 (18.87)
Current smoker	4205 (11.07)	55,954 (11.13)
BMI (kg/m^2^), mean(sd)	27.93 (4.95)	27.43 (4.80)
Physical activity, minutes/day, mean(sd)	75.50 (97.52)	74.23 (96.19)
Poor overall health rating	2412 (6.35)	22,780 (4.56)
Using cholesterol medication	8195 (21.57)	86,907 (17.29)
Using blood pressure medication	9610 (25.30)	104,024 (20.70)
Diabetes	2680 (7.05)	26,552 (5.28)
Heart diseases	3003 (7.91)	29,166 (5.80)
Breastfed as baby	21,051 (55.41)	277,596 (55.39)
Coffee consumption, cups/day, mean(sd)	2.02 (2.06)	2.01 (2.02)
Tea consumption, cups/day, mean(sd)	3.43 (2.73)	3.40 (2.69)
Oily fish consumption, servings/day, mean(sd)	0.16 (0.15)	0.16 (0.15)
Processed meat, servings/day, mean(sd)	0.22 (0.20)	0.21 (0.20)
Red meat, servings/day, mean(sd)	0.30 (0.21)	0.30 (0.21)
Fruit (fresh/dried), servings/day, mean(sd)	3.04 (2.60)	3.05 (2.62)
Vegetables (cooked/raw), servings/day, mean(sd)	0.82 (0.55)	0.81 (0.56)

* Data drawn from baseline (2006–2010). Values are numbers (%) unless stated otherwise.

**Table 2 nutrients-13-02114-t002:** Adjusted * OR (95% CI) of having a positive COVID-19 test by nutritional factors.

Nutritional Factor	Model 1		Model 2		Model 3	
	OR (95%CI)	*p* **	OR (95%CI)	*p* **	OR (95%CI)	*p* **
**Coffee, Cups/Day**						
None or <1 cup	Reference		Reference		Reference	
1 cup	0.90 (0.83, 0.97)	0.007	0.90 (0.83, 0.98)	0.015	0.93 (0.86, 1.01)	0.106
2–3 cups	0.89 (0.83, 0.95)	0.001	0.90 (0.83, 0.96)	0.003	0.92 (0.85, 0.99)	0.021
≥4 cups	0.92 (0.85, 0.996)	0.040	0.92 (0.84, 0.999)	0.047	0.91 (0.83, 0.99)	0.025
**Tea, cups/day**						
None or <1 cup	Reference		Reference		Reference	
1 cup	0.92 (0.82, 1.03)	0.157	0.93 (0.82, 1.04)	0.204	0.94 (0.84, 1.06)	0.319
2–3 cups	0.93 (0.85, 1.01)	0.074	0.93 (0.85, 1.01)	0.078	0.92 (0.84, 1.01)	0.069
≥4 cups	1.00 (0.92, 1.08)	0.941	0.98 (0.90, 1.06)	0.543	0.96 (0.88, 1.05)	0.347
**Oily Fish,** **Servings/Day**						
Quartile 1 (0–<0.07)	Reference		Reference		Reference	
Quartile 2 (0.07–<0.14)	0.93 (0.85, 1.02)	0.102	0.94 (0.86, 1.03)	0.183	0.97 (0.88, 1.06)	0.482
Quartiles 3 and 4 (≥0.14)	0.95 (0.87, 1.04)	0.244	0.98 (0.90, 1.07)	0.654	1.00 (0.91, 1.09)	0.967
**Processed Meat, Servings/Day**						
Quartile 1 (0–<0.07)	Reference		Reference		Reference	
Quartile 2 (0.07–<0.14)	1.02 (0.91, 1.14)	0.699	1.05 (0.93, 1.19)	0.410	1.04 (0.92, 1. 17)	0.547
Quartile 3 (0.14–<0.43)	1.07 (0.96, 1.20)	0.248	1.09 (0.97, 1.24)	0.155	1.07 (0.94, 1.21)	0.314
Quartile 4 (≥0.43)	1.12 (1.00, 1.25)	0.053	1.14 (1.01, 1.29)	0.036	1.12 (0.98, 1.26)	0.091
**Red Meat,** **Servings/Day**						
Quartile 1 (0–<0.21)	Reference		Reference		Reference	
Quartile 2 (0.21–<0.28)	0.96 (0.89, 1.04)	0.344	0.95 (0.87, 1.04)	0.236	0.99 (0.90, 1.08)	0.811
Quartile 3 (0.28–<0.35)	1.01 (0.93, 1.11)	0.771	1.00 (0.90, 1.10)	0.948	1.01 (0.91, 1.11)	0.903
Quartile 4 (≥0.35)	0.99 (0.92, 1.08)	0.878	0.98 (0.89, 1.07)	0.600	0.98 (0.90, 1.08)	0.697
**Fruit (Fresh/Dried), Servings/Day**						
Quartile 1 (0–<1.00)	Reference		Reference		Reference	
Quartile 2 (1.00–<2.25)	1.02 (0.92, 1.12)	0.778	1.05 (0.95, 1.16)	0.376	1.06 (0.95, 1.17)	0.300
Quartile 3 (2.25–<4.00)	0.97 (0.87, 1.09)	0.622	1.02 (0.91, 1.14)	0.762	1.03 (0.91, 1.15)	0.679
Quartile 4 (≥4.00)	0.97 (0.88, 1.09)	0.635	1.03 (0.92, 1.15)	0.660	1.03 (0.92, 1.16)	0.596
**Vegetables (Cooked/Raw), Servings/Day**						
Quartile 1 (0–<0.50)	Reference		Reference		Reference	
Quartile 2 (0.50–<0.67)	0.92 (0.85, 0.99)	0.034	0.93 (0.85, 1.00)	0.060	0.96 (0.88, 1.04)	0.271
Quartile 3 (0.67–<1.00)	0.87 (0.79, 0.96)	0.006	0.88 (0.80, 0.98)	0.015	0.93 (0.84, 1.03)	0.186
Quartile 4 (≥1.00)	0.90 (0.83, 0.98)	0.015	0.92 (0.84, 0.998)	0.046	0.96 (0.88, 1.05)	0.337
**Breastfed as a Baby**						
No	Reference		Reference		Reference	
Yes	0.91 (0.85, 0.98)	0.010	0.91 (0.85, 0.98)	0.013	0.95 (0.88, 1.02)	0.125
Don’t know	0.97 (0.91, 1.07)	0.750	0.98 (0.90, 1.07)	0.696	0.99 (0.91, 1.08)	0.789

***** Model 1: Adjusted for Townsend deprivation index, baseline age, sex, race, education, income, employment status, home ownership, number of co-habitants, BMI level, smoking status, physical activity, self-rated health, cholesterol-lowering medication use, antihypertension medication use, history of diabetes, and history of cardiovascular disease. Individual diet factors assessed in separate models. Model 2: Adjusted for all covariates listed in Model 1, with all diet factors included in the model (i.e., mutual adjustment). Model 3: Adjusted for all covariates listed in Model 2 and COVID-19 exposure score. ** An α of 0.05 was used as the cutoff for significance.

## Data Availability

The data that support the findings of this study are available from the UK Biobank upon approved request.

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
