# Peer review of "Dietary Behaviors and Incident COVID-19 in the UK Biobank"

_nutrients, 2021, doi:10.3390/nu13062114_

Round 1

Reviewer 1 Report

General:

The article by Vu et al. examined the association between 8 diet behaviors measured in 2006-2010 and incident COVID-19 in 2020 using data from the UK Biobank. It offers a timely and interesting overview of dietary factors potentially influencing the susceptibility to COVID-19 infection. The article is well-written, clear, and concise. The authors also accounted for a comprehensive set of potential confounders, which renders this observational study of interest. However, this manuscript requires some improvements prior to publication. The authors should state more clearly that the risk of COVID-19 infection is highly related to socio-economic conditions (e.g., size of social networks, occupations, access to masks, etc.), factors that are also highly correlated to diet behaviors. Moreover, although the preventive role of balanced diet in the COVID-19 progression and outcomes seems evident, notably thanks to a stronger immune system, the association between diet and the risk of COVID-19 infection is less biologically evident. Authors mentioned both aspects shortly in the discussion, but these should be more emphasized and cited as main limitations of the study. This has impact on the conclusions, too. In addition, the authors should better explain how dietary intake was measured. Finally, the article has some typos.

For your information, I had no access to supplementary materiel while conducting the review.

Comments:

  1. Abstract: Well written. However, the conclusion needs to be attenuated, knowing the two main limitations of your study mentioned above (i.e., possible residual confounding, no strong biological hypothesis). See below further comments for conclusion.
  2. There are several typos that need to be amended:
    1. L.36, remove the comma after crisis.
    2. L.43, space is missing between COVID-19 and the references.
    3. L.64 the reference [14] is misplaced, it should probably be at the ned of the parenthesis).
    4. L.79 not dot after “exposure”, and so on.
  3. L.76-78: It is unclear if you included people or tests. Please clarify. And how did you treat data from people who were tested twice or more? I suggest that you add a specific chapter to “Studied population” in the methods section.
  4. Baseline Dietary Data: More information about dietary assessment methods should be given. What were the response options to describe consumption frequency of food items in the FFQ (e.g., 1x/day, 2x/day)? Were these options similar for all food items? Was the FFQ semi-quantitative? If so, how portion sizes were evaluated? Was the FFQ completed once or several times?
  5. Statistical analyses: Crude models (e.g., Models 0) should be presented, at least in supplementary tables.
  6. Statistical analyses: Clarify the cut-offs used to define statistical significance (only done for interaction analyses, L. 162). In my view, using Bonferroni correction (e.g., 0.05/8) is needed since you tested 8 dietary parameters (plus several models!). State also in Table 2 footnote, which cut-offs were used for statistical significance.
  7. L.126: “F-tests or χ2 tests were used to compare differences in baseline characteristics across participants’ age and race/ethnicity.” What does it refer to? Supplemental Tables 1 and 2?
  8. L.143: The sentence “Model 2 was similar to Model 1 but with diet behaviors assessed mutually” should be clarified. Did the authors include all 7 other dietary parameters in the models? If so, please briefly mention the consequences this could have in the interpretation of the results. This paper might be useful: Ibsen DB, et al. Food substitution models for nutritional epidemiology. Am J Clin Nutr. 2021. PMID: 33300036
  9. L.167 et Table 1: To for clarity, “participants in the analysis sample” or “UKB analysis sample” should be referred more clearly as people who were tested for COVID-19. Otherwise, it is unclear on which criteria they were selected when reading Table 1 alone.
  10. Figure 1: The map legend about colors for waves 1 and 2 are unclear. I have the feeling that on the same map two time periods are presented, which is odd. Figure1’s title is also unclear. Please mention the units and cut-off points used for low vs. high positive test rates. The mention “assessed April 4, 2021” also confuses the reading about the analyzed period for UK COVID-19 surveillance data.
  11. L.199: “[…] the overall UKB COVID-19 sample”. This has to be rephrased since not all participants had COVID-19. Maybe use the term “study sample”.
  12. L.213: Make clear that it refers to Models 4. Otherwise, Models 4 are never mentioned in the results although described in the methods. Suggestion: “[…] did not alter results (Models 4, data not shown).”
  13. Discussion: State more clearly that the risk of COVID-19 infection is highly related to socio-economic conditions (e.g., size of social networks, occupations, access to masks, etc.), factors that are also highly correlated to diet behaviors. Based on that the conclusions in the abstract and main text should be attenuated.
  14. Discussion: Elaborate on the biological hypothesis of the preventive role a balanced diet would have on the risk of COVID-19 infection because it is not so evident.
  15. L.221-222. Sentence to be reformulated to clarify the message.
  16. L.261-264: This sentence is unclear. In addition, this is likely that opposite results about cabbage are probably due to chance. Please rephrase the sentence.
  17. L.270. To be reformulated as follows: […] COVID-19 in UKB. However, red meat […]
  18. L.314-317. Please elaborate since the link between tea, cognitive function and COVID-19 is not evident at first reading.
  19. Conclusions: Restrict the conclusions to your results based on COVID-19. Suggestions
    1. “Our results support the hypothesis that nutritional factors may influence distinct aspects of the immune system, hence susceptibility to COVID-19.”
    2. Encouraging adherence to certain nutritional behaviors (e.g., increasing vegetable intake and reducing processed meat intake), […].”
  20. Conclusions: Knowing your limitations (i.e., possible residual confounding, no strong biological hypothesis), the conclusions should be attenuated. Suggestion:
    1. Encouraging adherence to certain nutritional behaviors (g., increasing vegetable intake and reducing processed meat intake), may be an additional tool to existing COVID-19 protection guidelines to limit the spread of this virus”.
  21. Conclusions: Please state that your results should be confirmed in other populations.

Reviewer 2 Report

This study examined the relationship between dietary behaviors and COVID-19 incidence in a large population, by linking the UK Biobank data and COVID-19 test results. The contents of results are highly novel and very interesting for the broad readers of this journal, I think. In this paper, authors have properly considered the effect of COVID-19 exposure opportunities that should be considered when examining risk factors for infectious diseases. In addition, the relationship between dietary behaviors and COVID-19 incidence has been scrutinized from various perspectives such as age, race, and each epidemic wave, and has been confirmed the robustness of the study results. As a result, coffee consumption, fruit and vegetable intake, and the being breast-fed experience reduced the risk of COVID-19 incidence by 10%. The preventive risk percentage may be small, but dietary behaviors are modifiable by each person. If COVID-19 incidence may be prevented even a little by conducting preferable dietary behaviors in addition to COVID-19 preventive measures such as masks and hand hygiene, the impact could be large at the population level. Because there are more than 168 million cumulative cases of COVID-19 worldwide, more than 16 million cases could have been prevented by conducting preferable dietary behaviors. I think the result is really interesting for the reader.

Author Response

Dear Reviewer,

We are writing in response to your comments regarding our manuscript titled, “Dietary Behaviors and Incident COVID-19 in the UK Biobank” (Manuscript ID: nutrients-1241695). We agree with you that although the preventive risk percentage may be small, the total/attributable risk may high since coffee consumption, processed meat consumption and breastfeeding as a baby are highly prevalent and actionable behaviors. We are pleased to learn that the result from our study can be really interesting for the readers.

Thank you for your positive comments.

Round 2

Reviewer 1 Report

The manuscript by Vu et al., already well-written, clear, and concise, has improved. No further comments. 

Of note, I had access to supplementary files for the second revision.